# PAC-Bayesian Domain Adaptation Bounds for Multiclass Learners

**Anthony Sicilia**[1]     **Katherine Atwell**[2]     **Malihe Alikhani**[1,2]     **Seong Jae Hwang**[3*]

[1]Intelligent Systems Program, University of Pittsburgh, Pittsburgh, Pennsylvania, USA
[2]Department of Computer Science, University of Pittsburgh, Pittsburgh, Pennsylvania, USA
[3]Department of Artificial Intelligence, Yonsei University, Seoul, South Korea

## Abstract

Multiclass neural networks are a common tool in modern unsupervised domain adaptation, yet an appropriate theoretical description for their non-uniform sample complexity is lacking in the adaptation literature. To fill this gap, we propose the first PAC-Bayesian adaptation bounds for multiclass learners. We facilitate practical use of our bounds by also proposing the first approximation techniques for the multiclass distribution divergences we consider. For divergences dependent on a Gibbs predictor, we propose additional PAC-Bayesian adaptation bounds which remove the need for inefficient Monte-Carlo estimation. Empirically, we test the efficacy of our proposed approximation techniques as well as some novel design-concepts which we include in our bounds. Finally, we apply our bounds to analyze a common adaptation algorithm that uses neural networks.

## 1 INTRODUCTION

Multiclass neural networks are frequently used in implementation of many unsupervised domain adaptation algorithms. For example, neural networks are often employed for invariant feature learning algorithms [Ganin and Lempitsky, 2015, Long et al., 2017, 2018, Zhang et al., 2019], importance weighting algorithms [Lipton et al., 2018], or combinations of both techniques [Tachet des Combes et al., 2020]. While most of these adaptation algorithms are motivated by theoretical bounds, recent literature has paid close attention to the assumptions and failure-cases of some techniques [Zhao et al., 2019, Wu et al., 2019, Johansson et al., 2019]. Namely, some learning algorithms *ignore* key terms in the adaptation bounds on which they are based, and as a result, may output solutions (i.e., learned models) that violate as-

---

*Corresponding Author

sumptions and are *guaranteed* to fail at the adaptation task [Zhao et al., 2019, Wu et al., 2019]. Still, the story here is not totally complete. In particular, there has not been much discussion of the non-uniform sample complexity of these modern adaptation algorithms. Sample complexity, in fact, contributes an additional "ignored" term in the theoretical bounds on which modern adaptation algorithms are based.

In this paper, we propose the first multiclass adaptation bounds which allow us to study this non-uniform sample complexity. Studying sample complexity is important to our understanding of adaptation algorithms because it describes how "data-hungry" an algorithm is. When this sample complexity is non-uniform across an algorithm's solution space, it allows us to study properties of a solution as a function of its "data-hunger." This is especially important for adaptation algorithms, which as mentioned, can inadvertently output poor solutions. Identifying a dynamic relationship between the properties of solutions and their non-uniform sample-complexity can provide insight on how to prevent these failure-cases in practice (e.g., by collecting sufficient data for an algorithm). Non-uniform sample complexity (rather than uniform complexity) can also help us to better quantify implicit regularization inherent to our algorithm [Dziugaite and Roy, 2017, Nagarajan and Kolter, 2019]. Accurately describing implicit regularization is especially important when using neural networks [Neyshabur et al., 2014, Neyshabur, 2017, Keskar et al., 2017, Zhang et al., 2017], since similar learning algorithms can lead to solutions with distinct generalization performance and implicit regularization is believed to be the cause of this phenomena.

Despite the importance of studying non-uniform sample complexity in modern adaptation contexts, we are not aware of any multiclass adaptation bounds with this capability. To fill this gap, we contribute the first PAC-Bayesian adaptation bound for multiclass learners (Thm. 2). While PAC-Bayesian bounds actually control error for *stochastic* models, we choose this framework for its demonstrated empirical accuracy in describing neural network sample complexity [Dziugaite and Roy, 2017, Zhou et al., 2018, Jiang et al.,

*Accepted for the 38th Conference on Uncertainty in Artificial Intelligence (UAI 2022).*

2019, Dziugaite et al., 2020, 2021, Pérez-Ortiz et al., 2021]. Compared to existing bounds, we design our proposals to be more sensitive to the solution output by our learning algorithm as well as the data sample available for estimating key quantities. The former is vital in studying non-uniform complexity of adaptation algorithms (as discussed), while the latter is important for facilitating empirical study. To make our bound useful in practice, we also propose the first approximation techniques for the divergence terms in our bound. In one case, this involves proposal of a novel surrogate for optimizing 01-loss (Thm. 4). In another, we show a standard technique for computing divergence fails to generalize to the mutliclass setting without additional constraints (Thm. 5). Working in the PAC-Bayesian framework, some divergences we study are also expressed as expectations with no known analytic solution. For these, we propose additional bounds (Thm. 6, Cor. 1) which allow us to avoid inefficient Monte-Carlo estimation by introduction of a new flatness assumption related to the well-known flat-minima hypothesis [Hochreiter and Schmidhuber, 1997]. To conclude, we conduct extensive empirical study of more than 12K models learned across 5 diverse adaptation datasets.

## 2 BACKGROUND

### 2.1 NOTATION AND ASSUMPTIONS

Consider the space $\mathcal{X} \times \mathcal{Y}$ for some finite $\mathcal{Y}$ with $|\mathcal{Y}| > 2$ unless otherwise noted. Colloquially, we call $\mathcal{X}$ the feature space and $\mathcal{Y}$ the label space. For a distribution $\mathbb{D}$ over $\mathcal{X} \times \mathcal{Y}$, we are interested in the risk functional $\mathbf{R}_{\mathbb{D}} : \mathcal{Y}^{\mathcal{X}} \to [0, 1]$

$$\mathbf{R}_{\mathbb{D}}(h) \stackrel{\text{def}}{=} \mathbf{Pr}(h(X) \neq Y); \qquad (X, Y) \sim \mathbb{D} \qquad (1)$$

applied to some hypothesis (i.e., model) $h \in \mathcal{H} \subseteq \mathcal{Y}^{\mathcal{X}}$. The risk functional $\mathbf{R}_{\mathbb{D}}$ precisely gives the error rate of the hypothesis $h$ when tasked with modelling the relationship between $\mathcal{X}$ and $\mathcal{Y}$ described by $\mathbb{D}$. In PAC-Bayes, we also consider the risk of stochastic (Gibbs) predictors. For a distribution $\mathbb{Q}$ over $\mathcal{H} \subseteq \mathcal{Y}^{\mathcal{X}}$, the Gibbs risk is the expectation

$$\mathbf{R}_{\mathbb{D}}(\mathbb{Q}) \stackrel{\text{def}}{=} \mathbf{E}[\mathbf{R}_{\mathbb{D}}(H)]; \qquad H \sim \mathbb{Q}. \qquad (2)$$

For neural networks, a common stochastic formulation is to sample weights from the distribution $\mathbb{Q}$ before inference – e.g., the Bayesian neural networks of Blundell et al. [2015].

Throughout this paper, we assume a source distribution $\mathbb{S}$ over $\mathcal{X} \times \mathcal{Y}$ and a target distribution $\mathbb{T}$ over $\mathcal{X} \times \mathcal{Y}$. We assume observation of an i.i.d. random sample $S \sim \mathbb{S}^n$ and an i.i.d. random sample $T_X \sim \mathbb{T}_X^m$ where the subscript $X$ denotes the $\mathcal{X}$-marginal of a distribution. In this context, an algorithm for the unsupervised adaptation problem we study is a function $(S, T_X) \mapsto h \in \mathcal{H}$. We are interested in bounds on $\mathbf{R}_{\mathbb{T}}(h)$ for such algorithms.

Interchangeably, we think of the sample $S$ as both a random variable with distribution $\mathbb{S}^n$ and a (random) distribution

itself, since any observation of a sample $S$ uniquely defines its own empirical distribution over $\mathcal{X} \times \mathcal{Y}$ by the pmf

$$(x, y) \mapsto n^{-1} \sum_{i=1}^{n} \mathbf{1}_{\{(x_i, y_i)\}}\{(x, y)\} \qquad (3)$$

where $\mathbf{1}$ is the indicator function. So, $\mathbf{R}_S$ is well-defined by this identification. $\mathbf{R}_S(\mathbb{Q}) = \mathbf{E}_{H \sim \mathbb{Q}}[\mathbf{R}_S(H)]$ is also defined – the observation of $S$ is used, not integrated out.

Finally, we also use distribution divergences based on the $\mathcal{H}$-divergence proposed by Ben-David et al. [2007]. This divergence is a specification of the $\mathcal{A}$-distance [Kifer et al., 2004] which relaxes the total variation distance by considering only a subset $\mathcal{A}$ of measurable sets when taking the supremum. In particular, the $\mathcal{H}$-divergence considers sets identifiable by a class $\mathcal{H} \subseteq \{0, 1\}^{\mathcal{X}}$

$$\mathbf{d}_{\mathcal{H}}(\mathbb{D}_1, \mathbb{D}_2) \stackrel{\text{def}}{=} \sup_{h \in \mathcal{H}} \left| \mathbf{E}[h(X_1)] - \mathbf{E}[h(X_2)] \right| \qquad (4)$$

where $X_i \sim \mathbb{D}_i$. While it is typically defined with a factor of 2, we omit this for convenience. Given a class $\mathcal{H} \subseteq \mathcal{Y}^{\mathcal{X}}$, we first study the $\mathcal{H}\Delta\mathcal{H}$-divergence based on the class

$$\mathcal{H}\Delta\mathcal{H} \stackrel{\text{def}}{=} \left\{ x \mapsto 1 - \mathbf{1}_{\{h(x)\}}\{h'(x)\} \mid (h, h') \in \mathcal{H}^2 \right\}. \qquad (5)$$

This is a multiclass generalization, which simplifies to the original binary definition of Ben-David et al. when $|\mathcal{Y}| = 2$.

### 2.2 SOME EXISTING ADAPTATION BOUNDS

In this section, we discuss two adaptation bounds. More detailed knowledge of these bounds will be useful later for comparison with our proposed bounds. First, we discuss the seminal uniform convergence bound proposed by Ben-David et al. [2007, 2010]. Second, we discuss a PAC-Bayesian bound proposed by Germain et al. [2020].

#### 2.2.1 Adaptation Based on Uniform Convergence

**Theorem 1.** *[Ben-David et al., 2010] Let $\mathcal{Y}$ be binary. For all $\delta > 0$, w.p. at least $1 - \delta$, for all $h \in \mathcal{H}$*

$$\mathbf{R}_{\mathbb{T}}(h) \leq \lambda + \mathbf{R}_S(h) + \mathbf{d}_{\mathcal{H}\Delta\mathcal{H}}(S_X, T_X)$$
$$+ 4\sqrt{\frac{4\nu \ln(2m) - \ln(\delta/4)}{m}} + 2\sqrt{\frac{8\nu \ln(em/\nu) - 2\ln(\delta/4)}{m}} \qquad (6)$$

*where $\lambda = \min_{\eta \in \mathcal{H}} \mathbf{R}_{\mathbb{S}}(\eta) + \mathbf{R}_{\mathbb{T}}(\eta)$ and $\nu = \text{VCDim}(\mathcal{H})$.*

The seminal result above is the standard adaptation bound on which many newer results are based. Still, this uniform convergence bound is not well-suited for every application. We discuss some limitations below.

**Uniform Sample Complexity** Simply put, uniform convergence is too conservative: it assigns the same sample complexity to each outcome of our learning algorithm, regardless of the solution quality. As discussed in Section 1, this prevents us from studying important properties of a model as a function of its sample complexity.

**Model-Independent Divergence**    In general, divergence is meant to characterize the similarity in feature distributions under the source $\mathbb{S}$ and the target $\mathbb{T}$. Similar to above, independence of the divergence $\mathbf{d}_{\mathcal{H}\Delta\mathcal{H}}$ and the model $h$ is overly conservative and makes this term insensitive to changes in the outcome of our learning algorithm. For example, when $\mathcal{H}$ is fixed, this divergence cannot distinguish between a random initialization and a carefully trained solution.

**Sample-Independent Adaptability**    The term $\lambda$ is often called the *adaptability*. It is a measure of similarity in the labeling functions of $\mathbb{S}$ and $\mathbb{T}$, characterizing the extent to which one hypothesis in $\mathcal{H}$ can do well on *both* of these distributions. When no such hypothesis exists, it is unclear how a learner could successfully adapt by minimizing risk on the source distribution [Ben-David et al., 2010]. Importantly, this term has been central to the theoretical discussion of failure-cases in widely used DA algorithms [Johansson et al., 2019, Zhao et al., 2019]. Meanwhile, estimation of $\lambda$ remains an under-studied research area [Redko et al., 2020]. One problem, which we observe, is independence of $\lambda$ from the samples $S$ and $T$. In particular, one cannot directly compute the population statistic $\lambda$ in typical circumstances. Instead, one might estimate using $\min_{\eta} \mathbf{R}_S(\eta) + \mathbf{R}_T(\eta)$, but this requires verifying generalization of a learned model $h^* \in \arg\min_{\eta} \mathbf{R}_S(\eta) + \mathbf{R}_T(\eta)$ using a holdout set or some other descriptor of generalization performance (e.g., such as a learning bound). This is undesirable when, as in this paper, we wish to study adaptability in an empirical context. As we show in later experiments (Section 4), the extra generalization requirement typically inflates our estimation of $\lambda$, and subsequently, mars the results we would like to interpret.

**Binary Label Space**    It is also important to note that this bound was designed for binary learners. Computation of the $\mathcal{H}\Delta\mathcal{H}$-divergence is the most concerning issue, since existing algorithms for computation rely on symmetry of $\mathcal{H}$ and ERM over the class $\mathcal{H}\Delta\mathcal{H}$. In Section 3.2, we discuss these issues in detail and present some solutions.

### 2.2.2  A PAC-Bayesian Bound for Binary Learners

We give Thm. A in Appendix A.2, which is one of the first PAC-Bayesian adaptation bounds. While Germain et al. [2013, 2016, 2020] propose other bounds, we focus on Thm. A because it is easiest to compare to the proposal of Ben-David et al. [2010]. While tailored to Thm. A, the weaknesses discussed below are generally applicable to other bounds of Germain et al.

**Benefits Compared to Thm. 1**    One benefit of Thm. A is that the divergence employed in this bound is **model-dependent** (rather than independent); namely, it depends on the Gibbs predictor $\mathbb{Q}$, whose target error we bound. As mentioned, model-independence is an overly conservative

quality and Germain et al. [2020] show this formally by proving their divergence actually lowerbounds $\mathbf{d}_{\mathcal{H}\Delta\mathcal{H}}$ for all $\mathbb{Q}$ and $\mathcal{H}$. Another primary benefit is that Thm. A employs a **non-uniform** sample complexity. Specifically, complexity is measured through a KL-divergence $\mathrm{KL}(\mathbb{Q} \,\|\, \mathbb{P})$, which explicitly depends on the outcome of the learning algorithm $\mathbb{Q}$. Simply put, a model is complex if it deviates much from our prior knowledge, which is captured in the prior $\mathbb{P}$.

**Weaknesses Shared with Thm. 1**    Despite its benefits over Thm. 1, Thm. A also shares some weaknesses. First, the proposed adaptability term is also **sample-independent**. Second, the bound is still designed for a **binary label space** $\mathcal{Y}$. Unlike the case of Thm. 1, it is not computation of the bound that causes concern, but the *validity* of the bound in mutliclass settings. In particular, the problem arises because the proof of Thm. A relies on a decomposition of the risk which assumes $|\mathcal{Y}| = 2$. This decomposition does not hold, in general, when $\mathcal{Y}$ is larger. In fact, Germain et al. [2020], themselves, observe Thm. A is not easily extended to multi-class settings, leaving the investigation of such PAC-Bayes bounds as an open problem. For some additional empirical study of Thm. A, see Appendix D.3.

### 2.3  OTHER RELATED WORKS

Besides those works discussed above, there are some additional works, which propose alternate theories of adaptation. Some theories of adaptation use distinct integral probability metrics in place of the $\mathcal{H}$-divergence [Redko et al., 2017, Shen et al., 2018, Johansson et al., 2019], while others have sought to generalize and modify the $\mathcal{H}$-divergence [Mansour et al., 2009, Kuroki et al., 2019, Zhang et al., 2019]. Meanwhile, others focus on assumptions distinct from small adaptability. These include covariate shift [Sugiyama et al., 2007, You et al., 2019], label shift [Lipton et al., 2018] and *generalized* label shift [Tachet des Combes et al., 2020]. The DA problem can also be modeled through causal graphs [Zhang et al., 2015, Magliacane et al., 2018] and some extensions to DA consider a meta-distribution over targets [Blanchard et al., 2021, Albuquerque et al., 2020, Deng et al., 2020]. Notably, most assumptions are untestable in practice and not many works consider such testing, even in controlled research settings where it might be possible. As we are aware, we are the first to use a sample-dependent adaptability, which improves estimation in empirical study.

In adaptation, PAC-Bayesian results are almost exclusively due to Germain et al. [2013, 2016, 2020]. Albeit, in transfer learning some work does exist [Li and Bilmes, 2007, McNamara and Balcan, 2017]. Most directly, our work employs the PAC-Bayes bound of Maurer [2004] in proofs as well as some techniques of Langford and Caruana [2001], Dziugaite and Roy [2017], and Pérez-Ortiz et al. [2021] in empirical study. Most notably, ours is the only PAC-Bayesian work to

propose multiclass adaptation bounds. A more in depth coverage of relevant literature – for adaptation and PAC-Bayes – is available in Appendix B.

# 3 PROPOSED BOUNDS

In this section, we give the proposed adaptation bounds for multiclass learners. We also provide novel algorithms for computing two multiclass divergence terms and compare these to existing approaches. Lastly, we give a second adaptation bound which removes the need for inefficient Monte-Carlo estimation of divergences dependent on a Gibbs predictor. Proof of all results is given in Appendix A.

## 3.1 A PAC-BAYESIAN ADAPTATION BOUND FOR MULTICLASS LEARNERS

We begin by introducing a model-dependent class of hypotheses similar to $\mathcal{H}\Delta\mathcal{H}$. Precisely, for $h \in \mathcal{H}$,

$$h\Delta\mathcal{H} \stackrel{\text{def}}{=} \{x \mapsto 1 - \mathbf{1}_{\{h(x)\}}\{h'(x)\} \mid h' \in \mathcal{H}\}. \quad (7)$$

With it, we propose to use the $h\Delta\mathcal{H}$-divergence $\mathbf{d}_{h\Delta\mathcal{H}}$ in our adaptation bounds. This divergence is a **model-dependent** extension of the $\mathcal{H}\Delta\mathcal{H}$-divergence, applicable in multiclass settings. It is easy to observe from the definitions that this new divergence lowerbounds the $\mathcal{H}\Delta\mathcal{H}$-divergence for all $\mathcal{H}$ and all $h$. Both Zhang et al. [2019] and Kuroki et al. [2019] study similar divergences for bounding 01-loss in the binary case, but we are first to use this divergence with non-uniform sample complexity, and also, the first to use this divergence in an adaptation bound for multiclass learners.[1] Our full proposal requires novel technique and theoretical study to compute this divergence (see Section 3.2).

Next, we give the proposed adaptation bound. As alluded, the bound has a number of notable features and we expand on these in comparison to Thms. 1 and A after its statement.

**Theorem 2.** *For any $\mathbb{P}$ over $\mathcal{H}$, all $\delta > 0$, w.p. at least $1-\delta$, for all $\mathbb{Q}$ over $\mathcal{H}$*

$$\mathbf{R}_{\mathbb{T}}(\mathbb{Q}) \leq \tilde{\lambda}_{S,T} + \mathbf{R}_S(\mathbb{Q}) + \mathbf{E}_{H \sim \mathbb{Q}}[\mathbf{d}_{\mathcal{C}_H}(S_X, T_X)]$$
$$+ \sqrt{\frac{\text{KL}(\mathbb{Q}||\mathbb{P}) + \ln\sqrt{4m} - \ln(\delta)}{2m}} \quad (8)$$

*where $\tilde{\lambda}_{S,T} = \min_{\eta \in \mathcal{H}} \mathbf{R}_S(\eta) + \mathbf{R}_T(\eta)$ and we may choose either $\mathcal{C}_h = \mathcal{H}\Delta\mathcal{H}$ for all $h$ as before or $\mathcal{C}_h = h\Delta\mathcal{H}$.*

**Comparison to Thms. 1 and A** By design, the bound proposed above resolves the weaknesses mentioned in Section 2.2. First and foremost, we remove the requirement that $\mathcal{Y}$ is binary. Second, we use a **non-uniform** notion of sample complexity; i.e., $\text{KL}(\mathbb{Q} \| \mathbb{P})$. Third, Thm. 2 allows for

---

[1]We discuss a multiclass proposal of Zhang et al. [2019] later, but it is based on margin loss and used for uniform convergence.

---

*either* a **model-dependent** or **model-independent** notion of data-distribution divergence. While model-independent divergences do have some weaknesses, we retain them in our bound since, as discussed later, they can be more efficient. Lastly, Thm. 2 employs a **sample-dependent** notion of adaptability. Compared to $\lambda$, the new adaptability $\tilde{\lambda}_{S,T}$ is the smallest error achievable on the *samples*. In research contexts wherein we assume access to target labels for purpose of studying our assumptions, we later show that this quantity is fairly easy to empirically bound.

## 3.2 APPROXIMATING MULTICLASS DIVERGENCE

### 3.2.1 The Multiclass $\mathcal{H}\Delta\mathcal{H}$-divergence

First, since the $\mathcal{H}\Delta\mathcal{H}$-divergence is model-independent, the expectation with respect to the Gibbs predictor $\mathbb{Q}$ simplifies significantly. In particular, if $\mathcal{C}_h = \mathcal{H}\Delta\mathcal{H}$, we have

$$\mathbf{E}_{H \sim \mathbb{Q}}[\mathbf{d}_{\mathcal{C}_H}(S_X, T_X)] = \mathbf{d}_{\mathcal{H}\Delta\mathcal{H}}(S_X, T_X). \quad (9)$$

Thus, computation of this divergence simplifies to computing the $\mathcal{H}\Delta\mathcal{H}$-divergence for multiclass learners.

**Summary of Approach** In general, we take inspiration from the proposal of Ben-David et al. [2010] who compute $\mathcal{H}\Delta\mathcal{H}$-divergence when models in $\mathcal{H}$ have binary output. Namely, we frame computation as minimization of error in a specific classification problem. To adapt this strategy to the multiclass setting, we do two primary things. First, we remove the assumption that $\mathcal{H}$ is symmetric. This is important for multiclass settings since we have no reason to believe $\mathcal{H}\Delta\mathcal{H}$ is typically symmetric. We replace the symmetry in $\mathcal{H}\Delta\mathcal{H}$ with a symmetry in our classification problems. Second, for score-based classifiers such as neural networks, we give an optimization procedure for approximating ERM over this class based on a surrogate loss function. As we are aware, this is the first algorithm for approximating ERM over $\mathcal{H}\Delta\mathcal{H}$ when models in $\mathcal{H}$ have multiclass output.

**Reduction to ERM**

**Theorem 3.** *Let $\mathcal{C} = \mathcal{H}\Delta\mathcal{H}$. Almost surely,*

$$\mathbf{d}_{\mathcal{C}}(S_X, T_X) = \max \left\{ \begin{array}{l} 1 - \min_{\varphi \in \mathcal{C}} \mathbf{R}_P(\varphi) + \mathbf{R}_Q(\varphi), \\ 1 - \min_{\varphi \in \mathcal{C}} \mathbf{R}_U(\varphi) + \mathbf{R}_V(\varphi) \end{array} \right\} \quad (10)$$

*where*

$$P = ((X_i, 1) \mid X_i \in S_X), \ Q = ((\tilde{X}_i, 0) \mid \tilde{X}_i \in T_X),$$
$$U = ((X_i, 0) \mid X_i \in S_X), \ V = ((\tilde{X}_i, 1) \mid \tilde{X}_i \in T_X). \quad (11)$$

Notice, pooled samples $P +_c Q$ and $U +_c V$ define binary classification problems ($+_c$ is concatenation). Namely, they represent an identification problem wherein the learner must

distinguish between the samples $S_X$ and $T_X$. To compute divergence as above, we need only select $\varphi$ to minimize the sum of class-conditional error rates for these problems. Even in simple cases, risk-minimization can be computationally hard [Shalev-Shwartz and Ben-David, 2014]. Thus, we instead select $\varphi$ by optimizing a surrogate loss.

**Approximate Minimization via Surrogate** WLOG, $\mathcal{Y} = \{1, \ldots, C\}$. We consider a score-based class $\mathcal{S}$ written

$$\mathcal{S} \stackrel{\text{def}}{=} \{\Psi_{\mathbf{f}} \mid \mathbf{f} \in \mathcal{F}\}; \quad \Psi_{\mathbf{f}}(x) \stackrel{\text{def}}{=} \arg\max_{\ell \in [C]} \mathbf{f}_\ell(x) \quad (12)$$

with $\mathcal{F} \subseteq \{\mathbf{f} \mid \mathbf{f}_\ell : \mathcal{X} \to \mathbb{R}, \ \ell \in [C]\}$ a class of scoring-functions. In case of ties, suppose $\arg\max$ returns the least label. Using the naïve definition in Eq. (5),

$$\mathcal{S}\Delta\mathcal{S} \stackrel{\text{def}}{=} \{x \mapsto 1 - \mathbf{1}_{\{\Psi_{\mathbf{f}}(x)\}}\{\Psi_{\mathbf{g}}(x)\} \mid (\mathbf{f}, \mathbf{g}) \in \mathcal{F}^2\}. \quad (13)$$

At first glance, it is unclear how to pick $\varphi \in \mathcal{S}\Delta\mathcal{S}$ to minimize error on a given dataset. So, in place of this obscure definition, the following result gives a surrogate loss which upperbounds the 01-loss on the original problem. Thus, we indirectly reduce the error by minimizing the surrogate.

**Theorem 4.** *Suppose* $\tau : \mathbb{R} \to \mathbb{R}_{\geq 0}$ *is differentiable and monotone increasing. Let* $\mathbf{A} = \left(\tau \circ \mathbf{g}(x)\right) \cdot \left(\tau \circ \mathbf{f}(x)^{\mathrm{T}}\right)$ *with* $\tau$ *applied element-wise and* $\mathbf{f}, \mathbf{g} \in \mathcal{F}$. *Set*

$$\begin{aligned} z(x) &\stackrel{\text{def}}{=} \max_{(j,k) \in [C]^2} \mathbf{A}_{jk} - \max_{i \in [C]} \mathbf{A}_{ii}, \\ \mathcal{L}(z, y) &\stackrel{\text{def}}{=} \ln(1 + \exp(-(2y - 1) \cdot z))/\ln(2). \end{aligned} \quad (14)$$

*Then, if* $\varphi(x) = 1 - \mathbf{1}_{\{\Psi_{\mathbf{f}}(x)\}}\{\Psi_{\mathbf{g}}(x)\}$, *we have*

$$\mathbf{R}_{\mathbb{D}}(\varphi) \leq \mathbf{E}_{(X,Y) \sim \mathbb{D}} \mathcal{L}(z(X), Y) \quad (15)$$

*for any distribution* $\mathbb{D}$ *s.t.* $\mathbf{f}(X)$ *has no repeated scores and* $\mathbf{g}(X)$ *has no repeated scores, almost surely.*[2]

We point out the log loss $\mathcal{L}(z, y)$ is differentiable with respect to $z$ and $z(x)$ is differentiable with respect to $\mathbf{f}$ and $\mathbf{g}$. In practice, functions in $\mathcal{F}$ – such as $\mathbf{f}$ and $\mathbf{g}$ – are typically differentiable with respect to a real-parameter vector, which also defines the function. For example, this is precisely the case for neural networks. In these contexts, since composition preserves differentiability, the output of the surrogate $\mathcal{L}$ is differentiable with respect to the real-parameter vector. So, the RHS of Eq. (15) may be approximately minimized using batch SGD. At this point, the proposed algorithm should be familiar to the typical practitioner. It is equivalent to the manner in which we usually optimize a neural network, except for the new surrogate $(\mathbf{f}, \mathbf{g}, x, y) \mapsto \mathcal{L}(z(x), y)$.

---

[2]This stipulation on $\mathbb{D}$ is not too strict. It only assumes ties in the scores of $\mathbf{f}$ or $\mathbf{g}$ are *very* unlikely, so these ties can be ignored.

### 3.2.2 The Multiclass $h\Delta\mathcal{H}$-divergence

When $\mathcal{C}_h = h\Delta\mathcal{H}$ the divergence term is model-dependent and the expectation with respect to the Gibbs predictor $\mathbb{Q}$ becomes a challenge. For neural networks, even the Gibbs risk $\mathbf{R}_S(\mathbb{Q})$ does not have a known analytic solution. Instead, it is common to approximate using Monte-Carlo sampling [Langford and Caruana, 2001, Dziugaite and Roy, 2017, Dziugaite et al., 2021, Pérez-Ortiz et al., 2021]. By Hoeffding's Inequality, w.p. at least $1 - \delta$, we approximate

$$\mathbf{E}_{H \sim \mathbb{Q}}[\mathbf{d}_{\mathcal{C}_H}(S_X, T_X)] \leq \frac{1}{k}\sum_{i=1}^{k} \mathbf{d}_{\mathcal{C}_{H_i}}(S_X, T_X) + \sqrt{\tfrac{\ln 2/\delta}{2k}} \quad (16)$$

where $(H_i)_{i=1}^{k} \sim \mathbb{Q}^k$. Using the RHS as an approximation, our computation reduces to computing $\mathbf{d}_{\mathcal{C}_h}$ for any deterministic $h \in \mathcal{H}$. Upon sampling from $\mathbb{Q}$, we can apply the algorithm for computing $\mathbf{d}_{\mathcal{C}_h}$ to each point in the sample. In light of this, the next part focuses on computing $\mathbf{d}_{\mathcal{C}_h}$ for deterministic $h$. We proceed as before, reducing computation to risk-minimization for a specific classification problem.

**Reduction to ERM with Constrained Labeling Function**

**Theorem 5.** *Almost surely, for all* $h \in \mathcal{H}$

$$\mathbf{d}_{\mathcal{C}_h}(S_X, T_X) = \max \begin{cases} 1 - \min\limits_{\substack{\varphi \in \mathcal{H}, \\ \bar{h} \in \Upsilon}} \mathbf{R}_P(\varphi) + \mathbf{R}_Q(\varphi), \\ 1 - \min\limits_{\substack{\varphi \in \mathcal{H}, \\ \bar{h} \in \Upsilon}} \mathbf{R}_U(\varphi) + \mathbf{R}_V(\varphi) \end{cases} \quad (17)$$

*where* $\mathcal{C}_h = h\Delta\mathcal{H}$ *and*

$$\begin{aligned} P(\bar{h}) &= \left((X_i, \bar{h}(X_i)) \mid X_i \in S_X\right), \\ Q &= \left((\tilde{X}_i, h(\tilde{X}_i)) \mid \tilde{X}_i \in T_X\right), \\ U &= \left((X_i, h(X_i)) \mid X_i \in S_X\right), \\ V(\bar{h}) &= \left((\tilde{X}_i, \bar{h}(\tilde{X}_i)) \mid \tilde{X}_i \in T_X\right) \end{aligned} \quad (18)$$

*and* $\Upsilon = \{\bar{h} \in \mathcal{Y}^{\mathcal{X}} \mid \bar{h}(x) \neq h(x), \ \forall x \in \mathcal{X}\}$.

As before, the result describes two classification problems. This time, the learner's goal is to agree with $h$ on one sample, while disagreeing with $h$ (in the way specified by $\bar{h}$) on the other sample. We minimize the class-conditional error rates by selecting from the class $\mathcal{H}$ used for the original prediction task, rather than $\mathcal{H}\Delta\mathcal{H}$. So, we make the obvious proposal: re-use whichever approximation technique we used to select $h$ in the first place. In our experiments, since $\mathcal{H}$ is a space of neural networks, we use batch SGD on an NLL loss.

**A Heuristic for Selecting from $\Upsilon$** Reduction to ERM in the multiclass setting also requires specification of $\bar{h} \in \Upsilon$. Specifically, $\bar{h}$ should aid in minimizing the class-conditional error rates. In our experiments, we found a simple strategy to be fairly effective. Namely, we specify $\bar{h}$ by always picking the label with the second-highest confidence according to $h$. So, $\bar{h}$ disagrees with $h$ on all of $\mathcal{X}$,

but does so in the "most reasonable" way according to the probabilities output by $h$. This approach uses the probabilities output by $h$ to rank similarity of labeling functions and supposes the most "similiar" labeling function in $\Upsilon$ will be easiest for another hypothesis in $\mathcal{H}$ to simultaneously learn. Mathematically, our solution satisfies

$$\bar{h} \in \arg\max_{v \in \Upsilon} \sum_{x \in \mathcal{X}} h_{v(x)}(x) \qquad (19)$$

where $h_\ell(x)$ is the score assigned to the label $\ell$. This heuristic is a practical solution that avoids search over $\Upsilon$, which will typically be unknown, unless we inefficiently enumerate using membership constraints. As we are aware, there is no known algorithm to efficiently select minimizers from $\Upsilon$ and $\mathcal{H}$, simultaneously, as called for by Thm. 5. Besides the heuristic, we leave this problem as future work.

**Comparison to Some Related Approaches** Considering a binary label space $\mathcal{Y}$, Kuroki et al. [2019] propose a similar algorithm. The multiclass setting we consider does require some differences, primarily, related to the distinct degrees of freedom in multiclass and binary classification. First, our proposal removes the requirement that $\mathcal{H}$ is symmetric since, as we are aware, this concept is not well-defined for hypotheses with multiclass output. Similar to before, we replace the symmetry required of $\mathcal{H}$ with symmetry in our classification problems. Second, in multiclass settings, the reduction strategy necessitates a new parameter to optimize: the labeling function $\bar{h}$. Besides our proposed heuristic for optimization, proof of this fact is *not* a straightforward extension of the work of Kuroki et al. [2019]. In fact, it requires a different proof-technique (see Appendix A.6). In a multiclass setting, Zhang et al. [2019] also propose an approach for approximating a mutliclass divergence. In general, our two techniques require different consideration because their divergence is based on a margin loss, rather than 01-loss. Notably, the multiclass bounds of Zhang et al. [2019] use uniform sample complexity, unlike our proposed non-uniform approach. Further, working directly with 01-loss, as we do, avoids any loosening of the bound via the margin penalty.

### 3.3 EFFICIENCY THROUGH FLAT-MINIMA

In full view of Section 3.2.2, the reader may rightfully be concerned about the efficiency of the proposed technique for approximating $\mathbf{E}_H[\mathbf{d}_{\mathcal{C}_H}(\cdot, \cdot)]$. In particular, the suggestion requires training $k$ distinct neural networks: one for each $H_i \sim \mathbb{Q}$. Typically, $k$ will be large – e.g., larger than 100 – to control the size of the upperbound in Eq. (16), so this is not computationally feasible for practical applications. This problem is not totally unique to the model-dependent $h\Delta\mathcal{H}$-divergence, either. Common invariant feature-learning algorithms – e.g., DANN [Ganin and Lempitsky, 2015] – actually modify the feature distribution over which the classifier learns. In these cases, even the $\mathcal{H}\Delta\mathcal{H}$-divergence becomes

dependent on the model [Johansson et al., 2019]. In particular, supposing every model $h \in \mathcal{H}$ is the composition $c_h \circ f_h$ of a classifier $c_h$ and a feature extractor $f_h$, the modified $\mathcal{H}\Delta\mathcal{H}$-divergence results from the following restriction

$$[\mathcal{H}\Delta\mathcal{H}]_h \stackrel{\text{def}}{=} \{1 - \mathbf{1}_{\{c_p \circ f_h(\cdot)\}}\{c_q \circ f_h(\cdot)\} \mid (p,q) \in \mathcal{H}^2\}. \quad (20)$$

A similar restriction can be defined for the class $h\Delta\mathcal{H}$

$$[h\Delta\mathcal{H}]_h \stackrel{\text{def}}{=} \{1 - \mathbf{1}_{\{c_h \circ f_h(\cdot)\}}\{c_q \circ f_h(\cdot)\} \mid q \in \mathcal{H}\}. \quad (21)$$

In both cases, due to the dependence on $h$, the expectation over $\mathbb{Q}$ cannot be avoided as in Section 3.2.1. To resolve this frequent issue, we propose a new adaptation bound, which relies on an assumption related to flatness of the 01-loss over a (weighted) region in parameter space defined by $\mathbb{Q}$. Flatness assumptions are not unusual in PAC-Bayes and we develop this connection next.

#### 3.3.1 Flat-Minima and PAC-Bayes

An SGD solution lies in a flat-minimum if its parameters are robust to perturbation: changing the parameters (slightly) does not change the trained network's already low error rate. To put it another way, all parameter configurations near the SGD solution have identically low error. So, flatness here is an absence of "elevation" in error as our model moves about some region of parameter space. Hochreiter and Schmidhuber [1997] first discussed "flatness" as it relates to neural network generalization, hypothesizing that models lying in a large flat-minimum generalize well. More recently, the idea has been validated empirically at large scale. In particular, notions of the sharpness of minima are often good empirical descriptors of an SGD-trained neural network's generalization performance [Jiang et al., 2019, Dziugaite et al., 2020]. The motivation for using PAC-Bayes bounds is very often based on the hypothesis that flat-minima generalize well. This is because PAC-Bayes bounds implicitly encode the existence of flat-minima [Neyshabur et al., 2017, Dziugaite and Roy, 2017]. In details, for a bound to be small for some predictor $\mathbb{Q}$, both its Gibbs risk and its KL-divergence with the prior $\mathbb{P}$ must be small. Because the prior $\mathbb{P}$ typically has some variance, we know $\mathbb{Q}$ should have variance too, or else the KL-divergence will be large. Further, the variance of $\mathbb{Q}$ ensures a region of non-zero probability away from the mean. Thus, if the Gibbs risk is also small, it is required that models in this region away from the mean all have identically low error (i.e., form a flat-minimum). Otherwise, the Gibbs risk would be inflated by probable parameter configurations with high error as illustrated in Figure 1. In this sense, PAC-Bayes bounds and flat-minima go hand-in-hand. If the former is small, we know the latter exists.[3]

---

[3]This argument fails for some pathological cases. It works best with unimodal continuous $\mathbb{Q}$; e.g., the Gaussians used in Section 4.

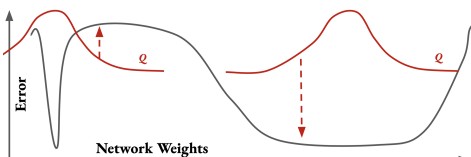

Figure 1: Informal illustration of flat-minimum (right) and sharp-minimum (left). For "flat" regions in parameter space, a unimodal Gibbs predictor $\mathbb{Q}$ with some variance has consistently low error across probable samples from $\mathbb{Q}$. Otherwise, when a region is "sharp", there is non-negligible likelihood of sampling a hypothesis from $\mathbb{Q}$ with high error. The expected error over $\mathbb{Q}$ is thus inflated by these likely regions of high error.

### 3.3.2 A More Efficient Adaptation Bound

**Definition 1.** *Let $\mathfrak{D}(\mathcal{H})$ be the space of distributions over $\mathcal{H}$ and fix a function $\mu : \mathfrak{D}(\mathcal{H}) \to \mathcal{H}$. The Gibbs predictor $\mathbb{Q}$ is $\rho$-flat on the distribution $\mathbb{D}$ if $|\mathbf{R}_{\mathbb{D}}(\mathbb{Q}) - \mathbf{R}_{\mathbb{D}}(\mu(\mathbb{Q}))| \leq \rho$.*

We call the function $\mu : \mathfrak{D}(\mathcal{H}) \to \mathcal{H}$ a *summary function* and call the image of $\mu$ a *summary*. When $\mathbb{Q}$ is implied, we typically abuse notation by writing $\mu = \mu(\mathbb{Q})$. Often, as in the definition above, we will refer to the "flatness" of a Gibbs predictor $\mathbb{Q}$ when it would be more precise to refer to the flatness of the (weighted) region in parameter space that this predictor defines (i.e., a around the summary $\mu$). In this sense, the above definition quantifies the flatness of a region in parameter space by the ability of the error in this region to be represented by a single hypothesis $\mu$ from that region. Intuitively, this echoes physical properties of flatness: a topographic map requires many more numbers to describe a mountainous terrain than a flat prairie. That is, each change in elevation for the mountainous terrain must be demarcated by individual numbers, while the flat prairie may only need one number to summarize the elevation. Similarly for the region around $\mu$ defined by the predictor $\mathbb{Q}$, a region is only "flat" if the error at $\mu$ is a good representative of the error across the whole region. Next, we give the proposed bound.

**Theorem 6.** *For any $\mathbb{P}$ over $\mathcal{H}$, all $\delta > 0$, w.p. at least $1 - \delta$, for all $\mathbb{Q}$ over $\mathcal{H}$ s.t. $\mathbb{Q}$ is $\rho_S$-flat on $S$ and $\rho_T$-flat on $T$*

$$\mathbf{R}_{\mathbb{T}}(\mathbb{Q}) \leq \rho + \tilde{\lambda}_{S,T} + \mathbf{R}_S(\mathbb{Q}) + \mathbf{d}_{\mathcal{C}_\mu}(S_X, T_X) \\ + \sqrt{\frac{\mathrm{KL}(\mathbb{Q}||\mathbb{P}) + \ln\sqrt{4m} - \ln(\delta)}{2m}} \quad (22)$$

*where $\mu$ is the summary of $\mathbb{Q}$, $\rho = \rho_S + \rho_T$, and $\mathcal{C}_\mu = \mu\Delta\mathcal{H}$.*

**Corollary 1.** *To study algorithms like DANN, we can instead choose $\mathcal{C}_\mu = [\mathcal{H}\Delta\mathcal{H}]_\mu$ or $\mathcal{C}_\mu = [\mu\Delta\mathcal{H}]_\mu$ in Thm. 6. The adaptability $\tilde{\lambda}_{S,T}$ is then dependent on $\mu$ as below*

$$\tilde{\lambda}_{S,T}^\mu = \min_{g \in \mathcal{H}} \left\{ \mathbf{R}_S(c_g \circ f_\mu) + \mathbf{R}_T(c_g \circ f_\mu) \right\}. \quad (23)$$

The main bound is identical to Thm. 2 except that we assume $\mathbb{Q}$ is flat on both $S$ and $T$, then use this assumption

to introduce a deterministic summary $\mu$ in the divergence. This deterministic summary replaces the expectation over $\mathbb{Q}$ whose estimation was inefficient, but the new cost is inflation of the bound by $\rho$. Unfortunately, similar to adaptability terms, we cannot expect to compute $\rho$ outside of controlled research contexts, since labels are required to estimate the flatness of $\mathbb{Q}$ on $T$ (according to Def. 1). Instead, for the bound to be practically useful, we propose to assume $\rho$ is small. This, for example, is often the suggestion when it comes to adaptability as well. Albeit, the caveats of carelessly making assumptions on adaptation problems should be noted [Zhao et al., 2019, Johansson et al., 2019].

We argue the assumption of small $\rho$ is not an overly strong (or careless) assumption to make. To begin with, PAC-Bayes bounds and flat-minima are already related. The only addition we make to the usual connection (see Section 3.3.1) is that flat regions *remain flat* when we transfer across data distributions (or, samples). Note, we do not even require the transferred region to remain a minimum, since the size of $\rho$ is only dictated by the *difference* in the Gibbs risk and the summary risk: the Gibbs risk can be high on $T$ as long as the summary risk is as well. Thus, if one is willing to accept the usual assumptions, then our additional assumption merely begs the question: *Do flat regions transfer?*

In the next section, empirically, we test this question along with the other proposals given in this text.

## 4 EXPERIMENTS

### 4.1 SETUP

**Datasets** We use a wide-array of datasets from vision and NLP: **Digits** [Ganin and Lempitsky, 2015], **PACS** [Li et al., 2017], **Office-Home** [Venkateswara et al., 2017], **Amazon Reviews** [Blitzer et al., 2007], and **Discourse** sense classification datasets [Prasad et al., 2008, Ramesh and Yu, 2010, Zeyrek et al., 2020]. For **Digits**, we use the image as feature, while for **Amazon Reviews**, we use uni-gram and bi-gram features. For other datasets, we use pre-trained ResNet-50 [He et al., 2016] or BERT [Devlin et al., 2019] features.

**Models** For **Digits**, we use a 4-layer CNN. For all other datasets, we use both a linear model and a 4-layer fully-connected network. For simplicity, our larger scale experiments use a simple adaptation algorithm (**SA**) which optimizes models to minimize risk on $S$. On **Digits**, we also study the **DANN** algorithm proposed by Ganin and Lempitsky [2015], modified to train Gibbs predictors with varied regularization using **PBB** [Pérez-Ortiz et al., 2021]. We pick **Digits**, specifically, because it exhibits shift in the marginal label distributions, which can cause **DANN** to fail [Zhao et al., 2019]. More training details are given in Appendix C.

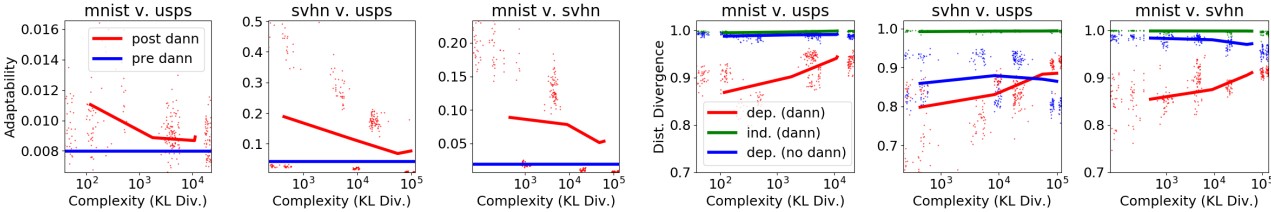

Figure 2: Adaptability (left) and dependent/independent divergences (right) for **DANN** on **Digits**. Solid line is median. Scatter describes unique $(S, T, \mathbb{Q})$, limited to 95% or more data to filter extreme values. $\mathbb{Q}$ is a multivariate Gaussian and $\mu$ is its mean.

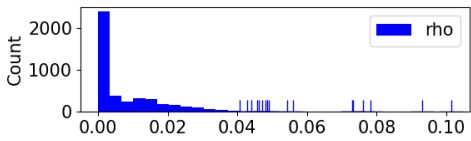

Figure 3: Histogram of $\rho$ estimates. Rug plot above 0.04 displays infrequent occurrences. Each datum describes unique $(S, T, \mathbb{Q})$. $\mathbb{Q}, \mu$ are defined as in Figure 2. See Appendix C.7 for details.

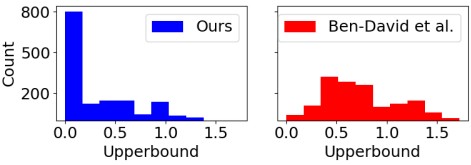

Figure 4: Sample-dependent (left) and independent adaptability. Each datum is for unique $(S, T, \mathcal{H})$. See Appendix C.5 for details.

**Experiments** The data points in our results are each individual experiments done on a source dataset $S$ and target dataset $T$ using a classifier $h$ or Gibbs predictor $\mathbb{Q}$. The pair $S$ and $T$ are taken from a set of data splits using the datasets discussed above (details in Appendix C). Across these splits, we consider various scenarios including: **single-source**, **multi-source**, and **within-distribution** adaptation (i.e., $\mathbb{S} = \mathbb{T}$) using multiple random data splits. On **Digits**, we also consider **natural shifts** (i.e., noise and rotation) and **unnatural shifts** (i.e., transfer to random data). In general, we restrict the pair $(S, T)$ to have a common label space.

### 4.2 RESULTS

**Sample-Dependent Adaptability** As mentioned, estimation of sample-independent adaptability (e.g., $\lambda$ in Thm. 1) requires verification of generalization. In particular, to estimate $\lambda$ one can learn $\eta \in \mathcal{H}$ which has small sum of risks over the observed samples $S$ and $T$. In our experiments, we do so using batch SGD on a weighted NLL loss – a common surrogate. Because $\lambda$ is a population statistic for $\mathbb{S}$ and $\mathbb{T}$, we cannot directly report the errors on $S$ and $T$ – this is incorrect, like using training error as a validation

Table 1: Correlation of $h$-dependent and -independent divergence with $|\mathbf{R}_h(S) - \mathbf{R}_h(T)|$. Columns delineate data subsets. Each datum describes unique $(S, T, h)$. See Appendix C.6 for details.

|            | All  | Digi. | Disc. | PACS+OH | Amaz. |
|------------|------|-------|-------|---------|-------|
| **model-ind.** | 0.54 | 0.15  | 0.70  | 0.41    | -0.05 |
| **model-dep.** | 0.58 | 0.23  | 0.78  | 0.14    | 0.41  |

metric. Instead, we should check the performance of $\eta$ on a heldout data subset (for example). This is the strategy we take in Figure 4, using Hoeffding's Inequality to produce a valid upperbound on $\lambda$. Comparably, estimating sample-dependent adaptability is much easier. By design, we *can* report error on the samples $S$ and $T$ used for training $\eta$. Doing so, produces a valid upperbound:

$$\forall \eta \in \mathcal{H} \ : \ \tilde{\lambda}_{S,T} \leq \mathbf{R}_S(\eta) + \mathbf{R}_T(\eta) \quad \text{(by definition).} \quad (24)$$

As is visible in Figure 4, this strategy for estimating $\tilde{\lambda}$ is much more effective than the sample-independent strategy in revealing important information. We see from the histogram of upperbounds on $\tilde{\lambda}$ that adaptability is very often small and concentrated near 0, although this is not always the case. Comparatively, upperbounds for $\lambda$ are spread out with notable mass at large values; we miss out on the interpretation that adaptability very often is small (as we might like to assume, in practice). In the rest of our discussion, all adaptability will be sample-dependent. Note, additional experiments on adaptability are available in Appendix D.1.

**Divergence and Approximation** In Table 1, we give results for our approximation techniques applied to the model-dependent $h\Delta\mathcal{H}$-divergence and the model-independent $\mathcal{H}\Delta\mathcal{H}$-divergence. The models used in these experiments are trained using **SA**. Since there is actually no ground-truth to compare too, we report performance of our approximations on a ranking task. That is, we compare our approximations to absolute difference in risks on the source and target and compute the Spearman rank correlation. According to our adaptation bounds, smaller divergence should predict smaller difference in risk and larger divergence should predict larger difference in risk as in the ranking task we study. Any effective approximation of divergence should also mimic this behavior, allowing us to conduct an indirect evaluation. In aggregate, we observe both divergences are

capable of ranking performance similarity on the source and target, which validates our approximations to some extent. For reference, a recent statistic designed for shift-detection [Rabanser et al., 2019] achieves correlation **0.29** on all data. We also observe the model-dependent divergence typically ranks "better" than the model-independent divergence. This, also, is to be expected according to our theory, since the model-independent divergence does not account for variation in $h$ and should thus perform worse. Overall, the nuanced agreement of our approximations with our theoretical expectations is suggestive that these techniques are effective.

**Do Flat Regions Transfer?** As noted, one stipulation of practical use for Thm. 6 is a small flatness value $\rho$. This is not unlike the common assumption that $\lambda$ is small and, as discussed, is related to the flat-minma hypothesis. To estimate $\rho$ and test our assumption, we select $\rho_S$ and $\rho_T$ to be the smallest values so that Def. 1 is satisfied on $S$ and $T$ using a Monte-Carlo estimate for the Gibbs Risk.[4] We train $\mathbb{Q}$ using a variant of **SA** based on the technique of Pérez-Ortiz et al. [2021]. Our results indicate $\rho$ is typically small as desired with mean **0.007** and SD **0.01** across 4K+ experiments. See Figure 3 for a visualization.

**Analysis of Assumptions after DANN** Our results in Figure 2 show an interesting relationship between the sample complexity of $\mathbb{Q}$ – as measured by $\mathrm{KL}(\mathbb{Q} \,||\, \mathbb{P})$ – and our assumption on adaptability. Namely, we can be more confident in the assumption $\tilde{\lambda}$ is small when the sample complexity of our solution increases. A similar observation holds for the flatness term $\rho$ (see Appendix D.4 Figure 8). Our analysis suggests **DANN** may be a data-hungry algorithm, since solutions with properties we desire have large sample complexity. The practical suggestion is to use large quantities of unlabeled data when applying **DANN**, which is reasonable since unlabeled data can be "cheap" to acquire.

**Analysis of Divergence after DANN** In Figure 2, according to the (more sensitive) model-dependent divergence, **DANN** reduces data-distribution divergence as it is designed to do. Still, it does not reduce divergence to the degree one might expect and, as the sample complexity of the solution increases, the gap between divergences – before and after **DANN** – begins to wane. This is interesting because it shows reduction of divergence and reduction of adaptability / flatness may be competing objectives. Further, this finding echoes theoretical hypotheses in recent literature [Zhao et al., 2019, Wu et al., 2019, Johansson et al., 2019], while also revealing the role of sample complexity in this story. To meet our assumptions when using **DANN**, we should use large amounts of unlabeled data and allow an unconstrained solution, but to ensure **DANN** reduces distribution diver-

gence significantly, we should instead constrain our solution to lower complexity (e.g., via regularization). Depending on problem context, there may be some optimum between these extremes, but in any case, these opposing relationships are an interesting take-away from the application of our theory.

# 5 CONCLUSION

In this work, we proposed the first adaptation bounds capable of studying the non-uniform sample complexity of adaptation algorithms using multiclass neural networks. Empirically, we validated the novel design-concepts in our adaptation bounds and showed our approximation techniques for some multiclass divergences were effective. In culmination, we applied our bounds to study sample complexity of a common domain-invariant learning algorithm. Our findings revealed unexpected relationships between sample complexity and important properties of the algorithm we studied. Code for reproducing our experiments is publicly available at `https://github.com/anthonysicilia/pacbayes-adaptation-UAI2022`.

Besides what has been done in this work, we also identify some areas of potential future work:

**Assumptions and Heuristics** As with previous adaptation bounds, the nature of the adaptation problem requires us to be imprecise in some cases. For one, we make a number of assumptions on adaptability and flatness. Also, our divergence computation does require some heuristics. While we study these imperfections empirically with promising results, we anticipate both shortcomings can be improved. In particular, restriction of scope to specific domains or hypothesis classes should reveal exploitable problem structure.

**Generalized Loss** While we have focused on multiclass learners in this work, a PAC-Bayesian adaptation bound for general learners (e.g., with bounded loss functions) remains an open-problem. Possibly, applying our strategies to the more general framework of Mansour et al. [2009] would be fruitful. Albeit, since algorithms for computing divergence have traditionally been loss-specific, we expect additional theoretical derivation to be required for each new loss.

### Acknowledgements

We thank the anonymous reviewers for helpful feedback.

S. Hwang was supported by Institute of Information & communications Technology Planning & Evaluation (IITP) grant funded by the Korea government (MSIT), Artificial Intelligence Graduate Program, Yonsei University (2020-0-01361-003), and the Yonsei University Research Fund of 2022 (2022-22-0131).

---

[4]A penalty based on Hoeffding Inequality could be added to this estimate to create a valid upperbound. We do not consider this since a penalty is also added if we use the strategy in Eq. (16).

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
