# OpenReview forum: "PAC-Bayesian Domain Adaptation Bounds for Multiclass Learners"
_auai.org/UAI/2022/Conference — UAI 2022 Oral_

### Official Review · Reviewer_P3S6 · 2022-04-01

**Q2(1) Originality/Novelty:** 3
**Q2(2) Significance/Impact:** 3
**Q2(3) Correctness/Technical Quality:** 4
**Q2(6) Clarity Of Writing:** 4
**Q6 Overall Score:** 7
**Q8 Confidence In Your Score:** 4

**Q1 Summary And Contributions:**

The paper present a pac-bayes bound for domain adaptation for multi-class classifiers. One of the differences from existing results is that the bound uses a sample-dependent adaptability, which should give tighter results compared to sample-independent adaptability. The bound also depends on a potentially, model dependent divergence term, measuring the discrepance between training distribution and target distribution, which requires novel techniques in order to estimate.

**Q2 Assessment Of The Paper:**

More detailed information regarding each of these aspects is given below:

**Q2(4) Quality Of Experiments (Optional):**

3: Good: The experimental evaluation is adequate, and the results convincingly support the main claims.

**Q2(5) Reproducibility:**

3: Good: Key resources (e.g., proofs, code, data) are available and key details (e.g., proofs, experimental setup) are sufficiently well-described for competent researchers to confidently reproduce the main results.

**Q3 Main Strengths:**

The result is novel, fairly demanding technically and addresses an important problem. The authors have combined ideas from the literature, with new tricks in order to deal with the general scenario of multiple classes with model dependent bounds.  Although the bound itself seems to be quite challenging to compute, the authors propose a method and offer a lot of discussion about it.

**Q4 Main Weakness:**

Computing the bound seems fairly tricky and requires certain heuristics; this seems to be the case for both Theorem 2 and Theorem 6, based on the flatness. I couldn't entirely follow the argumentation through flat minima.

**Q5 Detailed Comments To The Authors:**

* Concerning the relationship between $\rho$ and $KL(\mathbb{Q} | \mathbb{P})$, this could be also explained by the fact that a very concentrated $\mathbb{Q}$ would give high $KL$ and small $\rho$
*I think Lemma 1 in the appendix is automatic from the triangle inequality applied to the trivial metric.
*In equation (32) add brackets around argument of min, same in equation (51) and in general whenever there are min etc.
* In the proof of Lemma 4, I think "Binomial" should be "Bernoulli"
* Page 14 line -11 complements
* page 5, col 2, line -13, "similar"
* in the definition 1 how is $\mathbb{E}_{H\sim \mathbb{Q}} H$ defined? As I undestand $H$ will be a random function $\mathcal{X}\mapsto \mathcal{Y}$, where $\mathcal{Y}$ is the set of labels? even if we assign numerical labels the average itself the average may not be one of the labels

**Q7 Justification For Your Score:**

I think this is a good paper. Indeed computing the bound is tricky, but to my understanding, the paper makes significant progress wrt the literature. I checked most of the proofs and I couldn't find any missing details.

**Q9 Complying With Reviewing Instructions:**

1: Yes.

---

### Official Review · Reviewer_XTXJ · 2022-04-13

**Q2(1) Originality/Novelty:** 3
**Q2(2) Significance/Impact:** 3
**Q2(3) Correctness/Technical Quality:** 3
**Q2(6) Clarity Of Writing:** 3
**Q6 Overall Score:** 8
**Q8 Confidence In Your Score:** 3

**Q1 Summary And Contributions:**

The paper shows a new "adaptation" bound that connects risk of a model (or gibbs dist) on the source dist to its risk on the target distribution. The bound is sample-dependent and model-dependent both which makes it tighter than other bounds. One key contribution of the paper is that they their bound (or specifically upper bounds on it) can be reasonably estimated on the observed data.

**Q2 Assessment Of The Paper:**

More detailed information regarding each of these aspects is given below:

**Q2(4) Quality Of Experiments (Optional):**

3: Good: The experimental evaluation is adequate, and the results convincingly support the main claims.

**Q2(5) Reproducibility:**

3: Good: Key resources (e.g., proofs, code, data) are available and key details (e.g., proofs, experimental setup) are sufficiently well-described for competent researchers to confidently reproduce the main results.

**Q3 Main Strengths:**

The paper solves a very important problem and more or less gives a complete algorithm to estimate model and data-dependent bounds on adaptation + generalization. The theory is very interesting and the paper does a good job covering all the bases and giving algorithms to estimated a quantity or its upper bound, and after all is said and done the bounds are actually very indicative of the difference is S and T risks.

**Q4 Main Weakness:**

I don't see any strong concerns. One question I had was about was the PACS + OH result in table 1. Any reason the data-dependent one has that bad a correlation?

**Q5 Detailed Comments To The Authors:**

I don't see any strong concerns.

**Q7 Justification For Your Score:**

Seems like a good paper.

**Q9 Complying With Reviewing Instructions:**

1: Yes.

---

### Official Review · Reviewer_DEWY · 2022-04-15

**Q2(1) Originality/Novelty:** 3
**Q2(2) Significance/Impact:** 3
**Q2(3) Correctness/Technical Quality:** 3
**Q2(6) Clarity Of Writing:** 4
**Q6 Overall Score:** 8
**Q8 Confidence In Your Score:** 4

**Q1 Summary And Contributions:**

In the paper, the authors have proposed PAC-Bayesian adaptation bounds for multiclass learners. This work advances the state of the art, as earlier similar bounds are derived for binary label spaces, whereas, the proposed bounds are derived for the multiclass learners. Authors have derived approximation of these bounds that allows practical usage while removing the need for Monte-Carlo approximations. Authors have also provided novel algorithms to estimate the adaptation bounds.

**Q2 Assessment Of The Paper:**

More detailed information regarding each of these aspects is given below:

**Q2(4) Quality Of Experiments (Optional):**

3: Good: The experimental evaluation is adequate, and the results convincingly support the main claims.

**Q2(5) Reproducibility:**

3: Good: Key resources (e.g., proofs, code, data) are available and key details (e.g., proofs, experimental setup) are sufficiently well-described for competent researchers to confidently reproduce the main results.

**Q3 Main Strengths:**

1) Paper is very well written and provides a good justification of their motivation with a sufficient literature review.
2) The paper advances current state-of-the-art by extending PAC-Bayesian adaptation bounds of binary classifiers to multiclass learners.
3) Authors have provided mathematical proofs of their proposed adaptation bounds.
4) The proposed bounds can be used for models from different subfields of AI.

**Q4 Main Weakness:**

1) Authors have not yet made their codes public, although authors promise to do so once the paper is accepted.

**Q5 Detailed Comments To The Authors:**

This is a very well-written paper that derives PAC Bayesian adaptation bounds for multiclass learners. There are a few minor typos: for example "practioner".  Authors are requested to thoroughly review the paper for typographic and grammatical errors. It will be good if the authors could address the following queries:
1) Is it possible to extend the adaptation bounds to regression tasks?
2) Is there any effect of the choice of priors on the estimated bounds?
3) How are the correlations in Table 1 defined?

**Q7 Justification For Your Score:**

The overall score is based on a strong technical strength of the paper with sufficiently good mathematical explanations. Some efforts are required to revise the paper for minor typos. The overall impact of the paper will further improve if the authors make their codes publicly available.

**Q9 Complying With Reviewing Instructions:**

1: Yes.

---

### Decision · Program_Chairs · 2022-05-15

**Decision:**

Accept (Oral)

**Comment:**

Meta Review: All reviewers felt that the paper is worthy of publication. For preparing the camera-ready version, please ensure that you take the reviewers comments into account and make the paper as accessible as possible to a broader audience.